# Hypertension subtypes at high altitude in Peru: Analysis of the Demographic and Family Health Survey 2016–2019

Brando Ortiz-Saavedra[1], Elizbet S. Montes-Madariaga[1], Oscar Moreno-Loaiza[2], Carlos J. Toro-Huamanchumo[3,4]*

1 Universidad Nacional de San Agustín de Arequipa, Arequipa, Peru, 2 Institute of Biophysics Carlos Chagas Filho, Federal University of Rio de Janeiro, Rio de Janeiro, Brasil, 3 Unidad de Investigación para la Generación y Síntesis de Evidencias en Salud, Universidad San Ignacio de Loyola, Lima, Peru, 4 OBEMET Center for Obesity and Metabolic Health, Lima, Peru

* toro2993@gmail.com

## Abstract

### Background

The prevalence of hypertension in Peru has increased over the years. Approximately one third of the Peruvian population lives at high altitudes. This population presents particular physiological, genetic and environmental characteristics that could be related to the prevalence of hypertension and its subtypes.

### Objective

To assess the association between altitude and hypertension in the Peruvian population through an analysis of a nationally representative survey.

### Methods

We conducted a cross-sectional analysis of the Demographic and Family Health Survey for the period 2016–2019. We included 122,336 individuals aged 18 years and older. Hypertension was defined according to the JNC-7 guidelines. High-altitude location was defined as a residential cluster located above 2,500 meters above sea level. We utilized generalized linear models from the Poisson family with a log-link function to assess the magnitude of the association between high altitude and hypertension. Additionally, we employed multinomial regression models to analyze the association between high altitude and subtypes of hypertension, including isolated systolic hypertension (ISH), isolated diastolic hypertension (IDH), and systolic-diastolic hypertension (SDH).

### Results

In the adjusted Poisson regression model, we found that the prevalence of hypertension among participants living at high altitudes was lower compared to those living at low altitudes (aPR: 0.89; 95% CI: 0.86–0.93). In the adjusted multinomial regression model, we observed a lower prevalence rate of ISH among participants residing at high altitudes

e Informática del Perú (https://proyectos.inei.gob.
pe/microdatos/).

**Funding:** The author(s) received no specific
funding for this work.

**Competing interests:** The authors have declared
that no competing interests exist.

(aRPR: 0.68; 95% CI: 0.61–0.73) and a higher prevalence rate of IDH among participants
residing at high altitudes (aRPR: 1.60; 95% CI: 1.32–1.94).

## Conclusions

Residents at high altitudes in Peru have a lower prevalence rate of ISH and a higher prevalence rate of IDH compared to those living at low altitudes. Further studies are needed to determine the influence of other biological, environmental, and healthcare access factors on this relationship.

## Introduction

Hypertension is the leading risk factor for premature death [1]. In 2019, it accounted for 10.8 million deaths worldwide [2]. Likewise, hypertension is the most important risk factor for cardiovascular disease and mortality [1]. Various subtypes of this condition exist—such as isolated systolic hypertension (ISH), isolated diastolic hypertension (IDH), and systolic-diastolic hypertension (SDH) [3]. These different subtypes may have distinct prognostic implications and require individualized management [4]. Low- and middle-income countries have shown to have a high prevalence and low awareness of hypertension among patients [5]. Consequently, only about one-fourth of adults with hypertension receive treatment in these countries [6]. Peru, a middle-income South American country, has experienced a steady increase in the prevalence of hypertension in recent years (18.7% in 2015 to 20.6% in 2018), but awareness and treatment administration have not improved [6]. Likewise, from 1986 to 2015, hypertensive diseases rose from 15th to 10th place on the list of the main causes of mortality in the Peruvian population [7]. In the last 7 years, the mortality rate from hypertensive diseases increased, from 15.2 deaths per 100,000 inhabitants in 2017 to 38.5 deaths per 100,000 inhabitants in 2021 [8]. Therefore, studying potential local factors that could affect the disease's development is crucial for addressing this issue.

It is estimated that around 32% of the Peruvian population resides above 2,500 meters above sea level (masl) in the Andean highlands and is considered a high-altitude population [9]. The Andean population possesses distinct physiological and genetic features as a result of the adaptation process to chronic hypobaric hypoxia caused by exposure to high altitudes [10, 11]. The high-altitude environment is characterized by a decrease in barometric pressure and partial O2 pressure, lower temperatures, and higher ultraviolet solar radiation [12]. In the Andean population, several genes have been identified that show evidence of natural selection, including genes that may be involved in regulating blood pressure and cardiovascular function [13, 14]. These adaptive changes could be related to the prevalence of hypertension, its subtypes, and cardiovascular diseases.

Previous studies at altitude have shown an increase in the prevalence of hypertension with altitude exposure in Nepal [15]. Other studies have demonstrated a U-shaped relationship between altitude and hypertension in Tibet [16]. However, these findings cannot be applied to the Peruvian population, as the adaptation mechanisms differ in Andean residents [11]. Furthermore, the prevalence of hypertension is influenced by different factors specific to each population located at high altitude, such as age, sex, socioeconomic characteristics, race and culture [17]. Also, data on the distribution of hypertension subtypes at high altitudes are limited. Therefore, it is essential to assess the relationship between hypertension subtypes and altitude in the Peruvian population. The primary objective of this study was to assess the

association between altitude and hypertension subtypes in the Peruvian population through an analysis of a nationally representative survey.

## Materials and methods

### Study design, data source, and setting

We conducted a secondary cross-sectional analysis of data from the National Demographic and Family Health Survey (ENDES), administered by Peru's National Institute of Statistics and Informatics (INEI). The ENDES is an annual, nationally representative cross-sectional survey, conducted in line with the guidelines of The Demographic and Health Surveys Program (DHS) [18]. The survey used a two-stage, stratified, cluster random sampling approach, ensuring representation at national, departmental, and urban-rural levels [19]. Our study encompasses data from four consecutive annual surveys from 2016 to 2019. Peru has an approximate area of 1,285,215 km and is divided into three geographical regions: coast, mountains and jungle. The country is politically divided into 24 departments and the constitutional province of Callao. The Sierra begins approximately 500 meters above sea level, with the highest point being the snow-capped Huascaran, with a height of 6757 meters above sea level [20].

### Selection criteria

**Inclusion criteria.**   For the analysis, we included individuals who were 18 years of age or older.

**Exclusion criteria.**   We excluded pregnant participants, those with incomplete blood pressure data, and those exhibiting biologically implausible resting blood pressure measurements —specifically, systolic blood pressure (SBP) > 270 mmHg or < 70 mmHg, and diastolic blood pressure (DBP) > 150 mmHg or < 50 mmHg [21]. Additionally, we excluded participants with incomplete body mass index (BMI) data and those with biologically implausible BMI measurements, defined as BMI > 60 kg/m$^2$ or < 10 kg/m$^2$ [22].

### Variables

**Outcome.**   Hypertension was the primary dependent variable. Blood pressure was measured using a calibrated automatic sphygmomanometer (OMRON, Model HEM-713), with a measuring range of 0 to 299 mmHg and an accuracy of ±3 mmHg. Two types of cuffs were used depending on the participant's build: standard arm (220 to 320 mm) and larger arm (330 to 430 mm). Following general guidelines for blood pressure measurement [23], two readings were taken, separated by a two-minute interval. For the main analysis, we used the average of both readings. Hypertension was defined according to the "Seventh Report of the Joint National Committee on Prevention, Detection, Evaluation, and Treatment of high blood pressure: the JNC 7" [24], as either SBP $\geq$ 140 mmHg or DBP $\geq$ 90 mmHg or those participants who reported a previous diagnosis of hypertension. Three subtypes were defined: IDH as SBP < 140 mmHg and DBP $\geq$ 90 mmHg; ISH as SBP $\geq$ 140 mmHg and DBP < 90 mmHg; and SDH as SBP $\geq$ 140 mmHg and DBP $\geq$ 90 mm Hg [3].

Due to the white coat effect and the tendency for the first blood pressure reading to be higher than the second in epidemiological studies, we conducted a sensitivity analysis with three criteria to determine consistency between the two readings. We averaged both when the difference between the two SBP (or DBP) measurements was $\leq$5 mmHg. We used the second reading when the difference between the two SBP (or DBP) measurements was $\geq$6 mmHg, provided the first was higher. Finally, we excluded those measurements where the difference was $\geq$6 mmHg and the second was higher than the first [6].

**Exposure.** Exposure to high altitude was defined as residential location above 2,500 masl, which was measured for each cluster where the participant's home was located [25, 26]. Sensitivity analyses were also performed considering various altitude ranges (<500, 500–2,499, 2,500–3,499, ≥3,500 masl) [27].

**Covariates.** We included other covariates such as age (18–44, 45–54, 55–64, 65–74, ≥75) [28], sex (male, female), BMI (underweight-normal, overweight, obese), daily smoking status (yes, no), self-reported diabetes (yes, no), level of completed education (none, primary, secondary, higher), area of residence (urban, rural), and household wealth index (poorest, poor, middle, wealthy, wealthiest) [29].

**Bias.** The joining, manipulation, and curation of the data were performed with R software [30]. Two researchers independently downloaded and merged the databases. Likewise, to identify each of the variables, the dictionaries published by the INEI [19] were used and the recommendations of the DHS program methodological guide were also followed [18].

## Statistical analysis

Data analyses were conducted in R software (version 4.0.2) and STATA (version 14, College Station, Texas 77845, USA). In STATA software we performed the analysis using the *svy* and *subpop* commands. For all analyses we specified the sample design characteristics and weighting factors. A p-value of <0.05 was considered statistically significant in all analyses. Descriptive analyses were carried out using weighted proportions with their respective 95% confidence intervals and means with standard errors depending on the variable type. Bivariate analysis used Rao-Scott Chi-square tests for statistically significant differences between categorical variables, and design-based t-tests for numerical variables. We opted to use generalized linear models from the Poisson family and a long-link function to assess the association between high altitude and hypertension. Both crude and adjusted Prevalence Ratios (PR) with their respective 95% confidence intervals were reported. For the association analysis between high altitude and hypertension subtypes, multinomial regression models were used. We reported both crude and adjusted Relative Prevalence Ratios (RPR) with their respective 95% CIs. Additionally, stratified analyses were conducted by sex and area of residence as potential effect modifiers. Multicollinearity of independent variables was assessed through the generalized variance inflation factor (GVIF), using a cut-off point of <10 to rule out multicollinearity.

## Ethics

Data from the ENDES surveys are collected and published by INEI. The data are publicly available online (https://proyectos.inei.gob.pe/microdatos/). Ethical approval was not required as this is a secondary analysis of publicly available data, and the procedures and surveys of the DHS Program previously received ethical approval from the ICF Institutional Review Board (IRB). The ICF IRB ensures that the survey complies with the regulations of the U.S. Department of Health and Human Services for the protection of human subjects, while the host country's IRB ensures compliance with local laws and regulations [31].

## Results

### General characteristics of the study population

During the period from 2016 to 2019, a total of 132,917 individuals aged 15 and over were surveyed. After applying selection criteria, 10,581 participants (8.0%) were excluded, resulting in a total of 122,336 participants aged 18 and over. A summary of participant selection can be found in Fig 1.

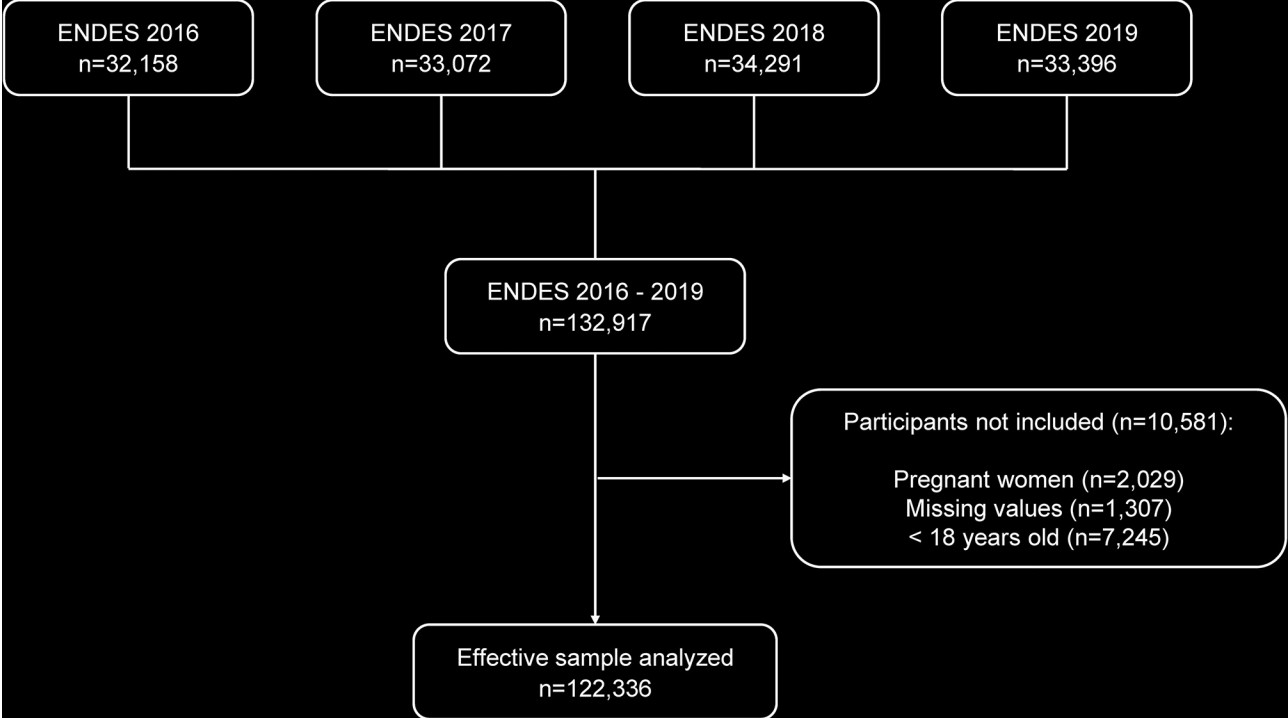

**Fig 1. Participant selection flowchart.**

The general characteristics of the participants are displayed in Tables 1 and 2. The majority of participants were women (51.1%). The proportion of adults aged 18 to 44 was 59.1%, and most people lived in urban areas (76.8%). The largest proportion of participants were married or cohabiting (66.6%), and higher education was the most common level of education (32.9%). The mean systolic and diastolic blood pressures were 122.5 (95% CI: 122.4–122.7) and 72.3 (95% CI: 72.2–72.4) mmHg, respectively. The largest proportion of participants were over-weight (41.0%), and 3.6% of respondents self-reported having diabetes. The prevalence of hypertension was 20.3% (Table 1).

## Characteristics according to altitude

Regarding inhabitants at high altitudes, 54.1% resided in rural areas, 35.6% had no formal education, and 47.4% fell into the quintile with the highest poverty. In contrast, for residents at low altitudes, only 14.6% resided in rural areas, 17.5% had no formal education, and just 12.4% belonged to the poorest quintile. For high-altitude residents, 14.4% had obesity and 1.8% self-reported having diabetes, whereas for low-altitude residents, 26.7% had obesity and 4.2% self-reported having diabetes. The prevalence of hypertension for residents at high and low altitudes was 17.0% and 21.2%, respectively. Finally, the prevalence of ISH, IDH and SDH at high altitudes was 6.2%, 1.1%, and 2.2%, respectively (Table 1).

## Characteristics according to hypertension subtypes

The percentage of ISH was higher among adults over 75 years old (35.4%). The percentage of IDH and SDH was higher among adults aged 45–54. The prevalence of all three hypertension subtypes was higher in males and in participants living in urban areas. The 17.8% of

**Table 1. Characteristics of the study participants according to altitude (n = 122,336).**

| | All | Low Altitude | High Altitude | p |
|---|---|---|---|---|
| | % (95% CI) | (< 2,500 masl) | (≥ 2,500 masl) | |
| | | % (95% CI) | % (95% CI) | |
| **Age** | | | | **<0.001** |
| 18–44 | 59.1 (58.6–59.5) | 59.8 (59.2–60.3) | 56.4 (55.6–57.2) | |
| 45–54 | 16.7 (16.4–17.1) | 16.6 (16.2–17.1) | 17.1 (16.5–17.7) | |
| 55–64 | 11.4 (11.1–11.7) | 11.5 (11.1–11.8) | 11.3 (10.9–11.8) | |
| 65–74 | 7.5 (7.2–7.7) | 7.2 (6.9–7.5) | 8.5 (8.1–8.9) | |
| ≥ 75 | 5.3 (5.1–5.6) | 4.9 (4.7–5.2) | 6.7 (6.3–7.1) | |
| **Sex** | | | | <0.001 |
| Female | 51.1 (50.6–51.5) | 50.6 (50.1–51.2) | 52.6 (51.8–53.4) | |
| Male | 48.9 (48.5–49.4) | 49.4 (48.8–49.9) | 47.4 (46.6–48.2) | |
| Area of residence | | | | <0.001 |
| Urban | 76.8 (76.2–77.3) | 85.4 (84.7–86.0) | 45.9 (44.3–47.5) | |
| Rural | 23.2 (22.7–23.8) | 14.6 (14.0–15.3) | 54.1 (52.5–55.7) | |
| Level of completed education | | | | <0.001 |
| No formal education | 21.4 (21.0–21.8) | 17.5 (17.0–17.9) | 35.6 (34.7–36.5) | |
| Primary | 17.9 (17.5–18.2) | 17.5 (17.1–18.0) | 19.1 (18.5–19.7) | |
| Secondary | 27.8 (27.4–28.2) | 29.2 (28.7–29.7) | 22.8 (22.1–23.5) | |
| Higher | 32.9 (32.4–33.4) | 35.8 (35.2–36.5) | 22.5 (21.7–23.3) | |
| Household wealth index | | | | <0.001 |
| Poorest | 20.0 (19.6–20.5) | 12.4 (11.9–12.9) | 47.4 (46.0–48.9) | |
| Poor | 20.9 (20.4–21.3) | 19.5 (19.0–20.0) | 25.7 (24.8–26.7) | |
| Middle | 20.3 (19.8–20.7) | 22.2 (21.6–22.7) | 13.5 (12.8–14.2) | |
| Wealthy | 19.7 (19.3–20.2) | 22.9 (22.3–23.4) | 8.6 (8.0–9.1) | |
| Wealthiest | 19.1 (18.5–19.7) | 23.0 (22.4–23.8) | 4.8 (4.3–5.2) | |
| SBP (mmHg)[a] | 122.5 (122.4–122.7) | 123.2 (123.0–123.4) | 120.3 (120.0–120.5) | <0.001 |
| DBP (mmHg)[a] | 72.3 (72.2–72.4) | 72.3 (72.2–72.4) | 72.5 (72.3–72.6) | 0.13 |
| BMI | | | | <0.001 |
| Underweight / Normal | 34.9 (34.5–35.4) | 31.4 (30.9–31.9) | 47.6 (46.7–48.4) | |
| Overweight | 41.0 (40.6–41.5) | 41.9 (41.3–42.4) | 38.0 (37.3–38.7) | |
| Obesity | 24.1 (23.6–24.5) | 26.7 (26.2–27.2) | 14.4 (13.9–15.0) | |
| Daily smoking status | 1.8 (1.7–1.9) | 2.2 (2.0–2.3) | 0.5 (0.4–0.6) | <0.001 |
| Self-reported diabetes | 3.6 (3.5–3.8) | 4.2 (3.9–4.4) | 1.8 (1.6–2.0) | <0.001 |
| Hypertension | 20.3 (19.9–20.7) | 21.2 (20.7–21.7) | 17.0 (16.4–17.6) | <0.001 |
| Hypertension subtypes (n = 111,588) | | | | |
| ISH | 7.7 (7.5–8.0) | 8.2 (7.9–8.6) | 6.2 (5.8–6.6) | <0.001 |
| IDH | 1.1 (1.0–1.2) | 1.0 (0.9–1.1) | 1.1 (1.0–1.3) | |
| SDH | 2.8 (2.6–2.9) | 2.9 (2.7–3.1) | 2.2 (2.0–2.5) | |

CI: Confidence interval; masl: meters above sea level; SBP: Systolic blood pressure: DBP: Diastolic blood pressure; BMI: Body Mass Index; ISH: Isolated Systolic Hypertension; IDH: Isolated Diastolic Hypertension: SDH: Systolic-diastolic Hypertension.

[a] Mean (95% CI)

participants with self-reported diabetes had ISH, and 3.5% had SDH (Table 2). The percentage point difference (PPD) in hypertension between the wealthiest and poorest was +6.5, while for education level, the PPD was -17.3 between higher education and no education (S1 Table in S1 File).

**Table 2. Characteristics of the participants according to hypertension of subtypes (N = 111,588).**

| | Without Hypertension % (95% CI) | ISH % (95% CI) | IDH % (95% CI) | SDH % (95% CI) | p |
|---|---|---|---|---|---|
| Age | | | | | <0.001 |
| 18–44 | 94.2 (93.9–94.5) | 3.2 (3.0–3.4) | 1.1 (0.9–1.2) | 1.5 (1.4–1.7) | |
| 45–54 | 85.1 (84.1–86.1) | 8.1 (7.3–8.9) | 1.6 (1.3–2.0) | 5.2 (4.6–5.8) | |
| 55–64 | 78.7 (77.3–80.0) | 15.5 (14.4–16.7) | 0.8 (0.6–1.1) | 5.0 (4.3–5.8) | |
| 65–74 | 69.1 (67.2–71.0) | 25.5 (23.7–27.4) | 0.4 (0.2–0.8) | 5.0 (4.0–6.1) | |
| >75 | 61.7 (59.0–64.3) | 35.4 (32.8–38.1) | 0.2 (0.1–0.6) | 2.7 (2.1–3.5) | |
| Sex | | | | | <0.001 |
| Female | 92.4 (92.0–92.8) | 5.7 (5.4–6.0) | 0.6 (0.5–0.7) | 1.3 (1.2–1.5) | |
| Male | 84.5 (84.0–85.1) | 9.8 (9.3–10.2) | 1.5 (1.3–1.7) | 4.2 (3.9–4.5) | |
| Area of residence | | | | | <0.001 |
| Urban | 88.0 (87.6–88.5) | 7.9 (7.5–8.2) | 1.1 (1.0–1.3) | 3.0 (2.8–3.2) | |
| Rural | 89.8 (89.4–90.3) | 7.4 (7.0–7.8) | 0.8 (0.7–0.9) | 2.0 (1.8–2.2) | |
| Level of completed education | | | | | <0.001 |
| No formal education | 81.3 (80.5–82.1) | 15.1 (14.3–15.8) | 0.7 (0.5–0.8) | 2.9 (2.7–3.3) | |
| Primary | 90.4 (89.7–91.1) | 6.1 (5.5–6.6) | 1.0 (0.8–1.2) | 2.5 (2.2–2.9) | |
| Secondary | 89.4 (88.7–90.0) | 6.8 (6.3–7.3) | 1.0 (0.9–1.2) | 2.8 (2.5–3.2) | |
| Higher | 90.7 (90.2–91.3) | 5.3 (4.8–5.7) | 1.3 (1.1–1.5) | 2.7 (2.4–3.0) | |
| Household wealth index | | | | | <0.001 |
| Poorest | 89.1 (88.6–89.6) | 8.0 (7.5–8.4) | 0.8 (0.7–0.9) | 2.1 (1.9–2.3) | |
| Poor | 89.9 (89.3–90.5) | 6.4 (5.9–6.9) | 1.1 (0.9–1.3) | 2.6 (2.3–3.0) | |
| Middle | 89.2 (88.4–89.9) | 7.3 (6.7–7.9) | 1.0 (0.9–1.3) | 2.5 (2.2–2.9) | |
| Wealthy | 87.3 (86.4–88.1) | 8.4 (7.7–9.2) | 1.0 (0.8–1.3) | 3.3 (2.9–3.8) | |
| Wealthiest | 86.5 (85.5–87.5) | 8.9 (8.1–9.7) | 1.3 (1.0–1.6) | 3.3 (2.8–3.9) | |
| BMI | | | | | <0.001 |
| Underweight / Normal | 92.2 (91.8–92.6) | 6.1 (5.7–6.4) | 0.4 (0.3–0.5) | 1.3 (1.2–1.5) | |
| Overweight | 88.3 (87.7–88.8) | 7.8 (7.4–8.2) | 1.1 (0.9–1.2) | 2.8 (2.6–3.2) | |
| Obesity | 82.7 (81.8–83.6) | 10.4 (9.7–11.1) | 2.0 (1.8–2.4) | 4.9 (4.4–5.3) | |
| Daily smoking status | 84.7 (81.8–87.2) | 9.6 (7.7–11.9) | 1.5 (0.8–2.8) | 4.2 (2.9–6.2) | 0.013 |
| Self-reported diabetes | 77.8 (74.8–80.5) | 17.8 (15.3–20.6) | 0.9 (0.5–1.9) | 3.5 (2.4–4.9) | <0.001 |

CI: Confidence interval; ISH: Isolated Systolic Hypertension; IDH: Isolated Diastolic Hypertension: SDH: Systolic-diastolic Hypertension; BMI: Body Mass Index.

## Association between high altitude and hypertension

Table 3 shows both the crude and adjusted Poisson regression models. The prevalence of hypertension in participants living at high altitudes was 11% lower compared to those living at low altitudes (aPR: 0.89; 95% CI: 0.86–0.93). In the stratified analysis by area of residence, the negative association remained, with rural areas having lower prevalence compared to urban areas. The negative association also persisted in the stratified analysis by sex, with females having a lower prevalence compared to males.

The results of the marginal predictions for the prevalence of hypertension at different altitudes are presented in Fig 2. An inverse relationship was observed in the direction of association between hypertension prevalence and altitude. This relationship was maintained in the analyses stratified by sex and age.

## Association between high altitude and hypertension subtypes

Table 4 shows both the crude and adjusted multinomial regression models. ISH was negatively associated with high altitude (aRPR: 0.68; 95% CI: 0.61–0.73), while IDH was positively

**Table 3. Association between high altitude and hypertension.**

| | Crude Model | | | Adjusted Model † | | |
|---|---|---|---|---|---|---|
| | cPR | 95% CI | *p* | aPR | 95% CI | *p* |
| **In all population (N = 122,336)** | | | | | | |
| 0–2,500 masl | Ref. | | | Ref. | | |
| >2,500 masl | 0.80 | 0.77–0.84 | <0.001 | 0.89 | 0.86–0.93 | <0.001 |
| **By area of residence** | | | | | | |
| Urban (n = 80,327) | | | | | | |
| 0–2,500 masl | Ref. | | | Ref. | | |
| >2,500 masl | 0.78 | 0.73–0.82 | <0.001 | 0.91 | 0.86–0.96 | 0.001 |
| Rural (n = 42,009) | | | | | | |
| 0–2,500 masl | Ref. | | | Ref. | | |
| >2,500 masl | 0.94 | 0.88–0.99 | 0.048 | 0.86 | 0.81–0.91 | <0.001 |
| **By sex** | | | | | | |
| Female (n = 69,034) | | | | | | |
| 0–2,500 masl | Ref. | | | Ref. | | |
| >2,500 masl | 0.85 | 0.83–0.90 | <0.001 | 0.85 | 0.80–0.90 | <0.001 |
| Male (n = 53,302) | | | | | | |
| 0–2,500 masl | Ref. | | | Ref. | | |
| >2,500 masl | 0.77 | 0.72–0.82 | <0.001 | 0.92 | 0.87–0.98 | 0.006 |

cPR: crude Prevalence Ratio, aPR: adjusted Prevalence Ratio, CI: Confidence Interval, masl: meters above sea level; Ref.: Reference.

† Model adjusted for: age, sex, BMI, daily smoker status, self-reported diabetes, area of residence, level of completed education, household wealth index.

associated with high altitude (aRPR: 1.60; 95% CI: 1.32–1.94). In the stratified analysis by area of residence, the aRPR remained relatively similar. In the stratified analysis by sex, a higher coefficient for IDH (aRPR: 2.03; 95% CI: 1.55–2.69) and a lower coefficient for ISH (aRPR: 0.55; 95% CI: 0.48–0.63) were observed in females compared to the general analysis.

## Sensitivity analysis

Six sensitivity analyses were conducted. In the first and second analyses, the criterion for hypertension that avoids the white coat effect and the categorization for high altitude (≥2,500 masl) were considered (S2 and S3 Tables in S1 File). In the third and fourth analyses, the criterion for hypertension that used the average of both blood pressure measurements and the categorization of altitude into: <500, 500–2,499, 2,500–3,499, and ≥3,500 masl were considered (S4 and S5 Tables in S1 File). Finally, for the fifth and sixth analyses, the criterion for hypertension that avoids the white coat effect and the categorization of altitude into: <500, 500–2,499, 2,500–3,499, and ≥3,500 masl were considered (S6 and S7 Tables in S1 File). In all analyses, the significant associations persisted.

## Discussion

### Main findings

The primary objective of this study was to evaluate the association between high altitude and subtypes of hypertension in the Peruvian population through a secondary analysis of ENDES during the period 2016–2019. Using data from the 122,336 participants included, we found that the prevalence of hypertension at high altitude was 17.0%, while at low altitude it was 21.2%. In the adjusted multivariable analysis, high altitude was negatively and significantly

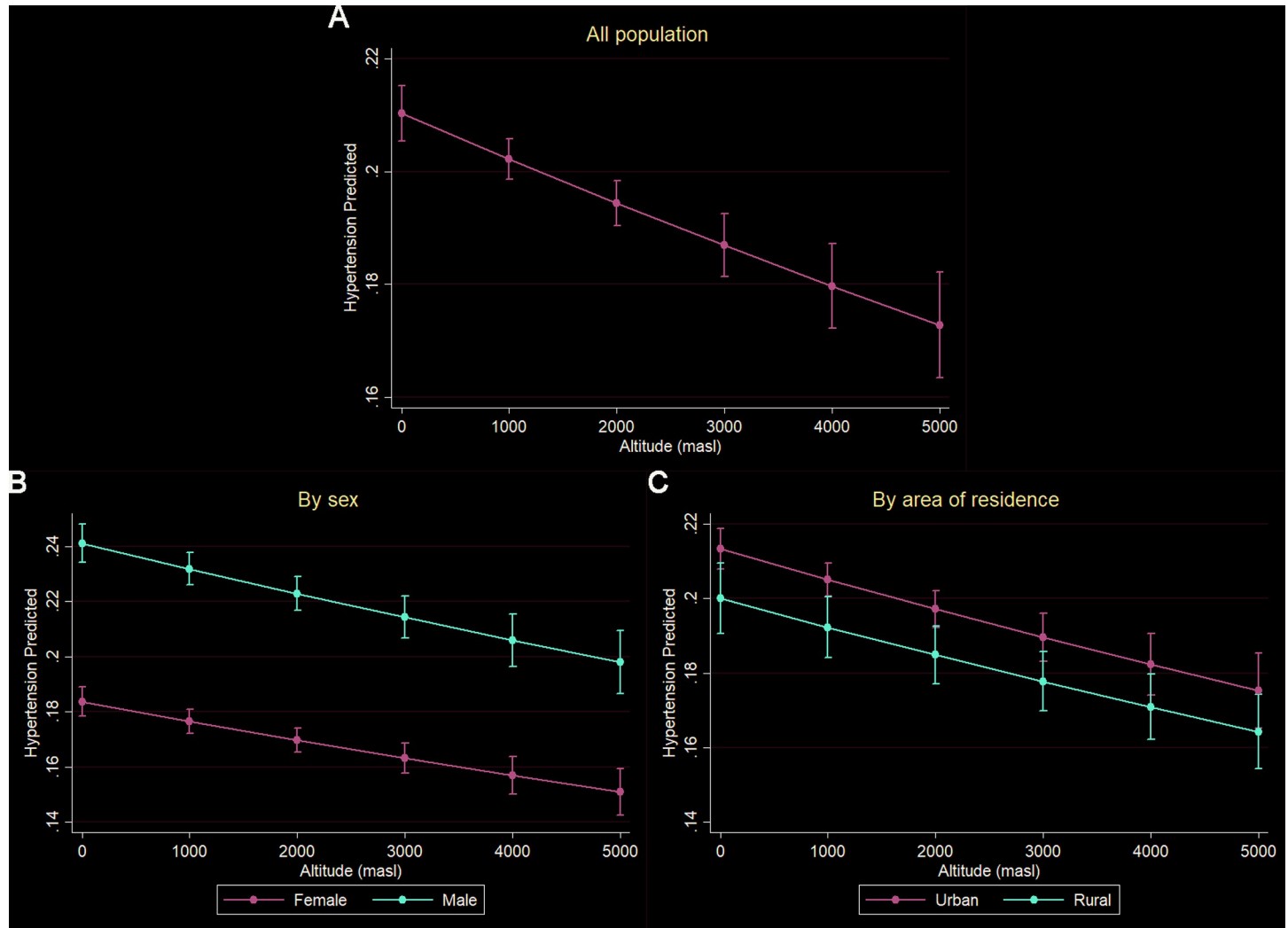

**Fig 2. Marginal effects of altitude on the prevalence of hypertension.** A) General population. B) Stratified by sex. C) Stratified by area of residence.

associated with hypertension. Similarly, systolic hypertension was negatively associated with high altitude, and diastolic hypertension was positively associated with high altitude.

## Comparison with previous studies

The negative association between ISH with high altitude and the positive association between IDH with high altitude found in the present study contrast with previous reports in Peru. Medina-Lezama et al. [32] conducted an analysis of the PREVENCION study on a stratified sample of a Peruvian Andean population, where they found that SDH was the predominant subtype of hypertension (41.8%), followed by ISH (29.2%) and IDH (28.2%). In the HIGH-CARE-ANDES Highlanders study, SDH was the most frequent subtype in participants with 24-hour (46%), daytime (52%), and office hypertension (55%), while ISH was the least frequent in all categories (<25%) [33]. Segura Vega et al. [34] conducted a secondary analysis of the TORNASOL I and II studies in people residing at high altitudes and found that the prevalence of IDH was 12.7% and that of ISH was 1.9%. However, the mentioned studies did not compare with a population residing at low altitudes. Regarding international studies, Gupta et al. [35]

**Table 4. Association between high altitude and hypertension subtypes.**

| | ISH | | | | IDH | | | | SDH | | | |
|---|---|---|---|---|---|---|---|---|---|---|---|---|
| | cRPR | 95% CI | aRPR | 95% CI | cRPR | 95% CI | aRPR | 95% CI | cRPR | 95% CI | aRPR | 95% CI |
| **In all population (N = 111,558)** | | | | | | | | | | | | |
| 0–2,500 masl | Ref. | | Ref. | | Ref. | | Ref. | | Ref. | | Ref. | |
| ≥2,500 masl | 0.73 | 0.67–0.79 | **0.68** | **0.61–0.73** | 1.08 | 0.90–1.28 | **1.60** | **1.32–1.94** | 0.75 | 0.66–0.85 | 0.98 | 0.86–1.12 |
| **By area of residence** | | | | | | | | | | | | |
| Urban (n = 73,062) | | | | | | | | | | | | |
| 0–2,500 masl | Ref. | | Ref. | | Ref. | | Ref. | | Ref. | | Ref. | |
| ≥2,500 masl | 0.59 | 0.52–0.68 | **0.64** | **0.55–0.73** | 1.26 | 1.01–1.58 | **1.57** | **1.24–2.00** | 0.76 | 0.64–0.90 | 0.91 | 0.76–1.09 |
| Rural (n = 38,526) | | | | | | | | | | | | |
| 0–2,500 masl | Ref. | | Ref. | | Ref. | | Ref. | | Ref. | | Ref. | |
| ≥2,500 masl | 0.91 | 0.81–1.01 | **0.68** | **0.60–0.78** | 1.35 | 0.99–1.82 | **1.66** | **1.22–2.26** | 1.09 | 0.90–1.33 | 1.09 | 0.89–1.34 |
| **By sex** | | | | | | | | | | | | |
| Female (n = 62,264) | | | | | | | | | | | | |
| 0–2,500 masl | Ref. | | Ref. | | Ref. | | Ref. | | Ref. | | Ref. | |
| ≥2,500 masl | 0.83 | 0.74–0.94 | **0.55** | **0.48–0.63** | 1.59 | 1.20–2.10 | **2.03** | **1.55–2.69** | 1.02 | 0.81–1.27 | 0.94 | 0.73–1.21 |
| Male (n = 49,324) | | | | | | | | | | | | |
| 0–2,500 masl | Ref. | | Ref. | | Ref. | | Ref. | | Ref. | | Ref. | |
| ≥2,500 masl | 0.68 | 0.61–0.76 | **0.71** | **0.62–0.80** | 0.91 | 0.73–1.14 | **1.38** | **1.07–1.78** | 0.69 | 0.59–0.80 | 0.92 | 0.78–1.08 |

RPR: Relative Prevalence Ratio; CI: Confidence Interval; ISH: Isolated Systolic Hypertension; IDH: Isolated Diastolic Hypertension: SDH: Systolic-diastolic Hypertension. masl: meters above sea level; Ref.: Reference

conducted an analysis of Nepal's 2016 Demographic and Health Survey, where they reported that the prevalence of ISH, IDH, and SDH was 3.3%, 7.5%, and 8.1%, respectively. However, no analyses were conducted to seek an association between the subtypes of hypertension and high altitude. Furthermore, Otsuka et al. reported that highland residents (3524 masl) had higher DBP compared to lowland residents (25 masl), although the study did not identify or analyze hypertension subtypes [36].

Concerning the negative association between high altitude and hypertension, it is consistent with what has been previously reported in studies conducted in Peru. Bernabé-Ortiz et. al. [37] carried out an analysis of the CRONICAS study (a prospective cohort type), where they found that people residing at high altitudes had a lower risk of hypertension. Meanwhile, in the TOR-NASOL II study [38], it was reported that the prevalence of hypertension was lower in communities located above 3,000 masl, compared to those located below. Likewise, Mendoza-Quispe et al. [39] reported that hypertension was lower in regions with altitudes higher than 2,500 and 3,500 m, compared to regions with altitudes lower than 1,500 m. However, there is a discrepancy with our findings in other studies conducted in populations residing at high altitudes. In studies conducted in Nepal, it was reported that the prevalence of hypertension and the mean SBP increased with altitude [15, 40]. Regarding the Tibetan population, it was found that altitude was U-shaped in its association with the likelihood of having hypertension [16]. Finally, in Ecuador, a non-linear J-shaped relationship was found between altitude and both SBP and DBP [41].

## Interpretation of results

ISH is mainly caused by increased stiffness and loss of distensibility of large elastic arteries, generally due to aging [42]. In general, arterial remodeling in highlanders may or may not involve an increase in arterial stiffness [11]. However, the adaptation mechanisms of large

elastic arteries in highlanders are still not clearly understood [43] and may differ between the Tibetan and Andean populations. Regarding IDH, it is more common in young adults and is generally a consequence of increased peripheral vascular resistance [44]. Sympathetic hyperactivity has been previously described in native Andean highlanders [45], which would lead to increased peripheral vasoconstriction [11] and thus elevated levels of diastolic blood pressure, consistent with our findings. Nevertheless, more studies are still needed in Andean highlanders to explain the mechanisms that modify the prevalence and course of ISH and IDH.

Concerning hypertension in general, the discrepancies found with other studies, mainly with those conducted in Asia, could be due to the fact that Andean and Tibetan populations had different patterns of physiological adaptation to hypobaric hypoxia at high altitudes [46]. Various characteristics of microcirculation differ in both populations, such as different levels of arterial oxygen content, oxygen saturation, hemoglobin concentration, and capillary network density [11]. These different adaptations suggest that Tibetans settled at high altitudes much earlier than Andean settlers [11], therefore the behavior of cardiovascular diseases could also be different.

It's important to mention that in the HIGHCARE-ANDES study conducted in Peru [33], masked hypertension was the most common phenotype of hypertension in populations residing at high altitudes, and the prevalence of hypertension tripled when a 24-hour ABPM (ambulatory blood pressure monitoring) was applied to participants with normotensive office measurements. Also, it is characteristic to find elevated levels of hemoglobin and erythrocytes in the Andean population, even in ranges of excessive erythrocytosis. This condition causes an increase in blood viscosity, hypervolemia, sympathetic activity, and oxidative stress, and as a consequence produces an elevation of blood pressure [33, 47]. Therefore, it's important that future studies consider these blood pressure measurement techniques for the correct diagnosis of hypertension in these populations.

In the analysis stratified by area of residence, we found that the prevalence of hypertension was lower in rural areas located at high altitudes, but not for ISH nor for IDH. Over the last 30 years, the prevalence of hypertension has increased in low- and middle-income countries, with a stronger trend in rural areas [48]. This could be due to the internal migration of young people from rural to urban areas and high levels of air pollution caused by the use of inefficient fuels at home, such as firewood, biomass, and coal [48]. Similarly, in developed countries like the United States, a higher prevalence of hypertension has been observed in rural areas [49]. This could be explained by a higher frequency of obesity, smoking, and physical inactivity in rural areas. Regarding the Andean region, there are different dietary patterns, characterized by low fat consumption [50]. In a community in the rural area of the Peruvian highlands, it was also found that energy derived from fat consumption was low [51]. However, in recent years in the rural area of the highlands, the consumption of sugars and cholesterol has increased, and the consumption of fruits and vegetables has decreased [51, 52]. These changes in diet could lead to an increase in the frequency of obesity and hypertension in the future. Regarding physical activity, inhabitants of the rural area of the Peruvian highlands burned more calories and did more aerobic exercises, generally as part of their daily agricultural and livestock activities [52, 53]. Therefore, these characteristic lifestyles of the high-altitude rural area could explain the low prevalence of hypertension; however, this could change in the coming years due to the changes found in dietary habits in this population.

## Relevance in public health

Our findings could have implications for Peru's public health policies related to hypertension, aiming to enhance primary care and lay the foundation for future lines of research. In 2015,

Peru's Ministry of Health approved the "Clinical Practice Guideline for the Diagnosis, Treatment, and Control of Hypertensive Disease" [54]. However, this guideline has not included special considerations for diagnosis and management in populations residing at high altitudes or for the specific subtypes of hypertension observed. For example, the guideline uses blood pressure value as the average of two measurements taken 2 minutes apart. Nonetheless, the utilization of blood pressure monitoring techniques would be necessary for individuals with elevated cardiovascular risk and excessive erythrocytosis, to identify masked hypertension and provide timely treatment.

Furthermore, since 2019, Peru has been part of the WHO's HEARTS Initiative. This initiative aims to integrate into established health services to promote the use of global best practices in the prevention and control of cardiovascular diseases such as hypertension [55]. By 2023, the goal is to implement at least 700 new healthcare facilities in Peru [56]. It would be of utmost importance for the initiative to be implemented in facilities situated at high altitudes and to develop specific guidelines for this population. These guidelines should include accurate identification of cardiovascular risk factors like dietary habits and physical activity, as well as special considerations for the correct diagnosis of hypertension.

Additionally, future longitudinal studies are needed to elucidate the pathophysiological mechanisms involved in the development of hypertension in individuals living at high altitudes. Important variables such as ethnicity, length of residence at high elevations, genomics, modifiable and non-modifiable risk factors unique to high-altitude areas, and correct hypertension diagnosis techniques should be considered.

### Limitations and strengths

Among the limitations of the present study is the fact that the study design does not allow us to establish causality between the exposure and outcome variables. Furthermore, due to the involvement of multiple interviewers, there may be inconsistencies in the techniques used for specific measurements. Recall bias could also be present when evaluating certain variables. Concerning the diagnosis of hypertension, the technique used for blood pressure measurement did not account for the masked hypertension phenotype [33], possibly leading to an underestimation of hypertension Prevalence. Additionaly, the ENDES considers a person a household resident if they slept in the household one day before the survey [57]. Consequently, many surveyed individuals may not be regular high-altitude residents, given the frequent interprovincial travel in Peru [58]. Another significant limitation is that the length of high-altitude residence is not considered, and many coastal individuals may relocate to the highlands and vice versa [58], due to various factors like employment. Lastly, the ancestry and ethnicity of high-altitude residents are not known, potentially affecting the results. Individuals with ancestors who have lived at high altitudes for multiple generations are more likely to have specific genes adapted to chronic hypoxia [46].

On the other hand, one of the study's strengths is that ENDES is a nationally representative survey, covering regional and urban-rural aspects, and adhering to the guidelines established by the DHS program. The survey team is well-trained, and various sensitivity analyses were conducted based on potential parameters that could alter the magnitude and directionality of the association between high altitude and hypertension.

### Conclusion

Residents of Peru's high-altitude regions have a lower prevalence of hypertension compared to those living at lower altitudes. Systolic hypertension has a lower prevalence rate in high-altitude inhabitants, while diastolic hypertension has a higher prevalence rate. Further studies are

needed to determine the influence of other biological, environmental, and healthcare accessibility factors on this relationship.

## Supporting information

**S1 File.**
(PDF)

## Acknowledgments

Brando Ortiz-Saavedra thanks Tech. Francis Saavedra for his continued support of this study.

## Author Contributions

**Conceptualization:** Brando Ortiz-Saavedra, Elizbet S. Montes-Madariaga.

**Data curation:** Brando Ortiz-Saavedra, Elizbet S. Montes-Madariaga, Carlos J. Toro-Huamanchumo.

**Formal analysis:** Brando Ortiz-Saavedra, Elizbet S. Montes-Madariaga, Oscar Moreno-Loaiza, Carlos J. Toro-Huamanchumo.

**Methodology:** Brando Ortiz-Saavedra, Elizbet S. Montes-Madariaga, Oscar Moreno-Loaiza, Carlos J. Toro-Huamanchumo.

**Project administration:** Brando Ortiz-Saavedra, Elizbet S. Montes-Madariaga, Oscar Moreno-Loaiza, Carlos J. Toro-Huamanchumo.

**Supervision:** Oscar Moreno-Loaiza, Carlos J. Toro-Huamanchumo.

**Validation:** Carlos J. Toro-Huamanchumo.

**Visualization:** Brando Ortiz-Saavedra, Elizbet S. Montes-Madariaga, Carlos J. Toro-Huamanchumo.

**Writing – original draft:** Brando Ortiz-Saavedra, Elizbet S. Montes-Madariaga, Oscar Moreno-Loaiza, Carlos J. Toro-Huamanchumo.

**Writing – review & editing:** Brando Ortiz-Saavedra, Elizbet S. Montes-Madariaga, Oscar Moreno-Loaiza, Carlos J. Toro-Huamanchumo.

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
