## [Decision Letter · Decision Letter 0]

12 Jan 2024

PONE-D-23-28046Hypertension subtypes at high altitude in Peru: Analysis of the Demographic and Family Health Survey 2016-2019PLOS ONE

Dear Dr. Toro-Huamanchumo,

Thank you for submitting your manuscript to PLOS ONE. After careful consideration, we feel that it has merit but does not fully meet PLOS ONE’s publication criteria as it currently stands. Therefore, we invite you to submit a revised version of the manuscript that addresses the points raised during the review process.

We look forward to receiving your revised manuscript.

Kind regards,

Esteban Ortiz-Prado

Academic Editor

PLOS ONE

Journal Requirements:

Additional Editor Comments:

Dear Authors,

Thank you for submitting your manuscript to our journal. After careful consideration and review by our experts in the field, including a detailed assessment by Reviewer 2, Dr. Eduardo Vasconez, we have decided that your manuscript can be accepted after minor revisions.

We appreciate the efforts you have made in conducting this significant research. The topic is of great importance and relevance, especially considering the unique environmental factors of high-altitude regions in Peru. Your findings contribute valuable insights to the field of medical geography and public health.

Based on the review, we request the following revisions to enhance the clarity and impact of your manuscript:

Summary Section: Please include a subtitle for 'Background' to provide a clearer structure.

Introduction Updates:

Update the data on global hypertension deaths, preferably using statistics from around 2020.

Clarify the increase in the prevalence of hypertension in Peru. Include specific figures or percentages to quantify this increase.

Incorporate data on the epidemiology of hypertension in Peru to provide context and relevance to your study.

Methods Refinement:

Add a 'Settings' section describing Peru's geographical divisions and altitude ranges.

Divide the 'Selection Criteria' into 'Inclusion Criteria' and 'Exclusion Criteria'.

Introduce a 'Bias' section explaining measures taken to minimize potential biases.

Clarify the rationale behind the chosen altitude classification, especially in comparison to the International Society of Mountain Medicine’s classification.

Results Consistency: Ensure that the altitude ranges discussed in the methodology align with those mentioned in the results.

Figure Quality: Improve the resolution and clarity of Figures 1 and 2.

Financial Disclosure: Relocate the 'Financial Disclosure Statement' to precede the reference list.

Your attention to these points will strengthen your manuscript and ensure its suitability for publication in our journal. We would appreciate receiving your revised manuscript by [insert deadline]. Please also include a detailed response to the reviewers' comments, outlining the changes made to the manuscript.

We look forward to your revised submission and believe that your research will make a valuable contribution to our journal.

Sincerely,

Reviewers' comments:

Reviewer's Responses to Questions

**Comments to the Author**

1. Is the manuscript technically sound, and do the data support the conclusions?

Reviewer #1: Yes

Reviewer #2: Yes

2. Has the statistical analysis been performed appropriately and rigorously? 

Reviewer #1: I Don't Know

Reviewer #2: Yes

3. Have the authors made all data underlying the findings in their manuscript fully available?

Reviewer #1: No

Reviewer #2: Yes

4. Is the manuscript presented in an intelligible fashion and written in standard English?

Reviewer #1: Yes

Reviewer #2: Yes

5. Review Comments to the Author

Reviewer #1: I appreciate the opportunity to review this interesting study “Hypertension subtypes at high altitude in Peru. Analysis of demographic and family health survey 2016-2019.”

Here is my few suggestion for consideration:-

Introduction section:- While few references have been mentioned regarding status of hypertension in high altitude regions of the world like south America, Nepal and Tibet, conspicuously missing is the reference on this subject in the Trans-Himalayan high altitude population of Indian part of Himalaya in Ladakh one of the highest inhabited regions in the Northernmost part of India. (BMJ Open 2015.) Reference 1. This study mentioned above concludes that like everywhere else in the world, hypertension prevalence in a high altitude population has multifactorial etiology. It shows that age, gender, socio-economic factors, culture, race and changing life style play a big role with the effect of high altitude itself on the prevalence of hypertension.

Method section:-

Being a secondary cross-sectional analysis of a data derived from national demographic and family health survey administered by Peru’s national institute of statistics and information, the data input and possibly analysis may not be questionable. However multiple investigators involved in data collection without regular supervision leaves considerable scope for methodological errors in the acquisition of the data. The authors though have admitted about this possibility in the section on limitation of the study.

Covariate section:- Most of the covariant like age strata , gender, core behavior, socio-economic status, including education, income, rural, urban are factored in. However varying dietary habits, and occupation important covariant for hypertension need to be factored into. Rural to urban migration also plays important role in hypertension through life style changes.

Result Section:- The primary objective of this study is evaluation of association between high altitude and subtypes of hypertension. To my knowledge there is no such specific study on the subtypes of hypertension at high altitude. However most of the previous studies seem to agree with their findings that prevalence of hypertension at high altitude is lower than that at lower altitude at least up-to age 70 years. What should have been evaluated in this study is the trajectory of blood pressure at high altitude beyond 70 years (70-80 years and ≥ 80 years.) A conspicuous observation in the Indian Himalayan population is that up-to age 70 years, altitude do-not play a significant role in prevalence of hypertension. However there is a steep rise in both systolic and diastolic blood pressure after 70 years. Probably it implies an accelerated aging after 70 years as it happens with chronic mountain sickness (CMS) prevalence with advancing age. (reference1). The negative association of ISH with altitude and positive association with IDH as seen in this study is in agreement with previous studies at high altitude. (K.Otsuka, T Norboo etal 2005 Reference 2).

Interpretation of results:- ISH is mainly caused by increased stiffness and loss of dispensability of large elastic arteries due to aging as postulated by authors is substantiated by other investigators. (K Otsuka and T.Norboo et al 2005 reference 3.) K Otsuka et al showed an extreme increase in Cardio-Ankle –Vascular Index (CAVI) with SDH in subjects with advanced age at high altitude which indicates increasing stiffness and loss of dispensability of large elastic arteries.

Statistic Part:- I regret my inability to evaluate this part and request editorial group to let this part be reviewed by an expert biostatistician.

The study is an original research, though based on secondary cross sectional analysis data. This study on subtypes of hypertension (ISH, IDH and SDH) has not been published elsewhere to my knowledge. The methodology of statistical analysis are described in detail, though I admit my inability to evaluate the data and leave it to the editorial staff to decide about it. Conclusions are presented in an appropriate fashion and is in conformity with the objective. The article is presented in intelligible fashion and is written in Standard English.

Reference:-

1) Tsering Norboo, Tsering Stobdan, Norboo Tsering, Norboo Angchuk, Phunsog Tsering, Iqbal Ahmed, and Tsewang Chorol, et al. Prevalence of hypertension at High Altitude: Cross sectional survey in Ladakh, Northern India 2007-2011.

2) K Otsuka, T. Norboo et al. “ Effect of aging in blood pressure in Leh, Ladakh, a high altitude (3524m) community by comparison with a Japanese town” ELSEVIER; Biomedicine and pharmacotherapy,. 59(2005)554-557.

3) K Otsuka , T.Norboo et al. ‘Chroneocological health watch of arterial stiffness and neuro-cardio- pulmonary function in elderly community at high altitude(3524m). ELSEVIER; Biomedicine and pharmacotherapy. 59 (20050558-567).

Reviewer #2: Dear authors, thank you for the opportunity to review your manuscript; I have some comments to improve the quality of your work

In the summary section include the subtitle background

Introduction

Line 59-60: The authors mention that in 2015 there were 8.5 million deaths, if possible please provide more updated data such as from 2020

Line 67-69: It is mentioned that Peru has presented an increase in the prevalence of hypertension. Can the authors mention how much this increase has been?

In the introduction section they do not mention anything about the epidemiology of hypertension in Peru, it would be good if they gave data on the situation of this disease in Peru.

Methods

Include the Settings section in which they describe the geography of Peru, for example, how many oans it is divided into and the height range of these areas.

In the Selection criteria section I suggest dividing it into inclusion criteria and exclusion criteria.

Include the bias section in which you explain what measures were taken to reduce the possibility of incurring some degree of bias.

Line 144-145: Why did the authors use that height classification? instead of the classification given by the International Society of Mountain Medicine (low altitude (<1,500 m), moderate altitude (1,500–2,500 m), high altitude (2,500–3,500 m) and very high altitude (3,500–5,500 m), or because they did not use the height of the 8 regions of Peru: Costa, Yunga, Quechua, Suni, Puna, Janca, Selva Alta and Selva Baja

Results

In the methodology I will mention the following height ranges (<500, 500-2,499, 2,500-3,499, ≥3,500 masl) but in the results they only talk about two ranges < 2,500 masl, ≥ 2,500 masl. Can the authors clarify this, already that they establish in their methodology does not coincide with the results section

Improves the image quality of both figure 1 and figure 2, which are a little blurry.

Move the FINANCIAL DISCLOSURE STATEMENT section so that it is located at the top of the list of references

6. PLOS authors have the option to publish the peer review history of their article (what does this mean?). If published, this will include your full peer review and any attached files.

Reviewer #1: **Yes: **Tsering Norboo

Reviewer #2: No

---

## [Author Response · Author response to Decision Letter 0]

23 Jan 2024

Hypertension subtypes at high altitude in Peru: Analysis of the Demographic and Family Health Survey 2016-2019

Reviewer #1:

Comment: Introduction section: While few references have been mentioned regarding status of hypertension in high altitude regions of the world like south America, Nepal and Tibet, conspicuously missing is the reference on this subject in the Trans-Himalayan high altitude population of Indian part of Himalaya in Ladakh one of the highest inhabited regions in the Northernmost part of India. (BMJ Open 2015.) Reference 1. This study mentioned above concludes that like everywhere else in the world, hypertension prevalence in a high altitude population has multifactorial etiology. It shows that age, gender, socio-economic factors, culture, race and changing life style play a big role with the effect of high altitude itself on the prevalence of hypertension.

Response: Thank you very much for the comment. We added this information in the introduction (line 96 with reference No. 17, of the version with track changes).

Comment: Covariate section: Most of the covariant like age strata, gender, core behavior, socio-economic status, including education, income, rural, urban are factored in. However varying dietary habits, and occupation important covariant for hypertension need to be factored into. Rural to urban migration also plays important role in hypertension through life style changes. 

Response: Thank you very much for observation. We also consider that it would be important to introduce these covariates in the analysis, however the survey does not have objective information on these variables. We selected covariates according to what was reported in previous studies and their availability in the survey.

Comment: Result Section: However most of the previous studies seem to agree with their findings that prevalence of hypertension at high altitude is lower than that at lower altitude at least up-to age 70 years. What should have been evaluated in this study is the trajectory of blood pressure at high altitude beyond 70 years (70-80 years and ≥ 80 years.) A conspicuous observation in the Indian Himalayan population is that up-to age 70 years, altitude do-not play a significant role in prevalence of hypertension. However, there is a steep rise in both systolic and diastolic blood pressure after 70 years. Probably it implies an accelerated aging after 70 years as it happens with chronic mountain sickness (CMS) prevalence with advancing age. (reference1). The negative association of ISH with altitude and positive association with IDH as seen in this study is in agreement with previous studies at high altitude. (K.Otsuka, T Norboo etal 2005 Reference 2). 

Response: Thank you very much for your comment. For the age covariate, we used the cutoffs described in the “2020 International Society of Hypertension Global Hypertension Practice Guidelines.” Likewise, we added and discussed reference No. 2 in the discussion section (line 332, Revised Manuscript with Track Changes).

Reviewer #2:

Comment: Summary Section: Please include a subtitle for 'Background' to provide a clearer structure.

Response: We added the "Background" section in the summary.

Comment: Introduction Updates: Update the data on global hypertension deaths, preferably using statistics from around 2020. 

Response: Thank you very much for your suggestion. We have updated the data.

Comment: Introduction Updates: Clarify the increase in the prevalence of hypertension in Peru. Include specific figures or percentages to quantify this increase.

Response: Thank you very much for your comment. We added the percentages (line 72 of the introduction section [Revised Manuscript with Track Changes]).

Comment: Introduction Updates: Incorporate data on the epidemiology of hypertension in Peru to provide context and relevance to your study.

Response: Thank you very much for your comment. We added the suggested information (lines 73 - 77 [Revised Manuscript with Track Changes]).

Comment: Add a 'Settings' section describing Peru's geographical divisions and altitude ranges.

Response: We added the “Settings” section along with the “Study Design and Data Source” section, aiming to provide better understanding to the reader.

Comment: Divide the 'Selection Criteria' into 'Inclusion Criteria' and 'Exclusion Criteria'.

Response: We carried out the suggested division.

Comment: Introduce a 'Bias' section explaining measures taken to minimize potential biases.

Response: Thanks a lot for the suggestion. We added the "Bias" section, and it is located on line 172 (Revised Manuscript with Track Changes).

Comment: Clarify the rationale behind the chosen altitude classification, especially in comparison to the International Society of Mountain Medicine’s classification.

Response: We thank the reviewer for highlighting the issue of high altitude classification. We have chosen 2500 meters above sea level (masl) as the cutoff for defining high altitude, as this is a widely recognized and practical benchmark in existing literature for studying the impact of high altitude on health. Furthermore, both the consensus on acute and chronic mountain sickness by the International Society of Mountain Medicine (ISMM) identify these conditions as occurring above 2500 masl (Roach RC et al., High Alt Med Biol 2018; Beall, Annu Rev Anthropol 2014). To clarify this, we have added the relevant references in the 'Variables: Exposure' section. Additionally, we conducted sensitivity analyses using other cutoff points (<500, 500-2,499, 2,500-3,499, ≥3,500 masl), as employed in a previous study with national data (Bernabe-Ortiz 2022, Glob Heart).

Comment: Results Consistency: Ensure that the altitude ranges discussed in the methodology align with those mentioned in the results.

Response: The results obtained with the cut-off point of 2500 masl are found in the manuscript, while the result of the sensitivity analyzes with other cut-off points (<500, 500-2,499, 2,500-3,499, ≥3,500 masl) are found in the file S1_File (supplementary material).

Comment: Figure Quality: Improve the resolution and clarity of Figures 1 and 2.

Response: We improved the quality of both figures.

Comment: Financial Disclosure: Relocate the 'Financial Disclosure Statement' to precede the reference list.

Response: Done.

---

## [Editor Report · Decision Letter 1]

28 Feb 2024

Hypertension subtypes at high altitude in Peru: Analysis of the Demographic and Family Health Survey 2016-2019

PONE-D-23-28046R1

Dear Dr. Toro-Huamanchumo,

We’re pleased to inform you that your manuscript has been judged scientifically suitable for publication and will be formally accepted for publication once it meets all outstanding technical requirements.

Kind regards,

Esteban Ortiz-Prado

Academic Editor

PLOS ONE

Additional Editor Comments (optional):

Dear authors,

As the invited editor for your manuscript titled "Hypertension subtypes at high altitude in Peru: Analysis of the Demographic and Family Health Survey 2016-2019," I have completed my final review of your revised submission.

I am pleased to inform you that I have no further comments or suggestions, and I am impressed by the thoroughness of your revisions and the effort you have put into addressing the feedback from the review process. Your dedication to enhancing the quality and impact of your work is evident and much appreciated.

Please look out for further communication from the journal regarding the next steps towards publication.

Warm regards,

Dr. Esteban Ortiz
---

## [Editor Report · Acceptance letter]

22 Mar 2024

PONE-D-23-28046R1 

PLOS ONE

Dear Dr. Toro-Huamanchumo, 

I'm pleased to inform you that your manuscript has been deemed suitable for publication in PLOS ONE. Congratulations! Your manuscript is now being handed over to our production team.

Kind regards, 

on behalf of

Dr. Esteban Ortiz-Prado 

Academic Editor

PLOS ONE